# Molecular Biology and Clinical Management of Esophageal Adenocarcinoma

**DOI:** 10.3390/cancers15225410

**Published:** 2023-11-14

**Authors:** Shulin Li, Sanne Johanna Maria Hoefnagel, Kausilia Krishnawatie Krishnadath

**Affiliations:** 1Center for Experimental and Molecular Medicine, Amsterdam University Medical Centers, University of Amsterdam, 1105 AZ Amsterdam, The Netherlands; s.li@amsterdamumc.nl; 2Cancer Center Amsterdam, 1081 HV Amsterdam, The Netherlands; 3Department of Internal Medicine, CWZ, 6532 SZ Nijmegen, The Netherlands; s.hoefnagel@cwz.nl; 4Department of Gastroenterology and Hepatology, Antwerp University Hospital, 2650 Edegem, Belgium; 5Laboratory of Experimental Medicine and Pediatrics, University of Antwerp, 2000 Antwerpen, Belgium

**Keywords:** esophageal adenocarcinoma, Barrett’s esophagus, gastroesophageal reflux disease, etiology, genetics, epigenetics, treatment

## Abstract

**Simple Summary:**

Esophageal adenocarcinoma (EAC) is one of the two main subtypes of esophageal cancer. EAC is a highly lethal disease with rising incidence in Western countries. EAC is associated with chronic gastroesophageal reflux disease and Barrett’s esophagus and mostly occurs in the distal esophagus. In the past decades, much effort has been made to improve treatment strategies, including regimens with chemoradiotherapy, targeted and immune therapies. Despite the multi-modal therapies, the survival of EAC patients has improved only marginally, and major breakthroughs in EAC treatment have not been achieved. We aim to summarize the literature on the comprehensive molecular landscape of EAC to elucidate factors underlying the EAC malignant behavior and poor outcomes. We discuss in detail the etiology, genetics, epigenetics and histology of EAC, as well as the currently employed therapies. Better knowledge about the molecular biology of EAC learned from this review may provide leads for developing novel therapies in the future.

**Abstract:**

Esophageal adenocarcinoma (EAC) is a highly lethal malignancy. Due to its rising incidence, EAC has become a severe health challenge in Western countries. Current treatment strategies are mainly chosen based on disease stage and clinical features, whereas the biological background is hardly considered. In this study, we performed a comprehensive review of existing studies and discussed how etiology, genetics and epigenetic characteristics, together with the tumor microenvironment, contribute to the malignant behavior and dismal prognosis of EAC. During the development of EAC, several intestinal-type proteins and signaling cascades are induced. The anti-inflammatory and immunosuppressive microenvironment is associated with poor survival. The accumulation of somatic mutations at the early phase and chromosomal structural rearrangements at relatively later time points contribute to the dynamic and heterogeneous genetic landscape of EAC. EAC is also characterized by frequent DNA methylation and dysregulation of microRNAs. We summarize the findings of dysregulations of specific cytokines, chemokines and immune cells in the tumor microenvironment and conclude that DNA methylation and microRNAs vary with each different phase of BE, LGD, HGD, early EAC and invasive EAC. Furthermore, we discuss the suitability of the currently employed therapies in the clinic and possible new therapies in the future. The development of targeted and immune therapies has been hampered by the heterogeneous genetic characteristics of EAC. In view of this, the up-to-date knowledge revealed by this work is absolutely important for future EAC studies and the discovery of new therapeutics.

## 1. Introduction

Esophageal cancer (EC) is predominantly divided into histological subtypes: esophageal adenocarcinoma (EAC) and esophageal squamous cell carcinoma (ESCC). Different from ESCC, which mostly occurs in the upper and middle esophagus, EAC is mainly (>90%) located in the lower esophagus in the proximity of the gastro-esophageal junction [1,2,3]. This review focuses on EAC. The incidence rate of EAC has been increasing in Western countries [4]. EAC is a highly aggressive cancer and known for rapid tumor dissemination leading to distant metastases, which is associated with poor survival [5]. Barrett’s esophagus (BE) induced by gastroesophageal reflux disease (GERD) is the only known precursor of EAC. EAC develops via a stepwise process, commonly in compliance with the well-known sequence of GERD/BE/BE with dysplasia/adenocarcinoma. Obesity and GERD are two main risk factors associated with BE and EAC [6,7,8,9]. Other risk factors include male sex, smoking and Caucasian ethnicity [10]. In the past decades, much effort has been made in identifying and monitoring BE patients via endoscopic surveillance in order to detect early progression to dysplasia and prevent the occurrence of EAC in a timely manner. Although the majority of individuals with BE (90–95%) under surveillance do not progress to EAC or are treated successfully in the case of dysplasia or mucosal cancers, a small portion of BE patients (5–10%) even in careful surveillance can rapidly develop EAC [11]. The lesions may be missed potentially because of a lack of visible endoscopic abnormalities and sampling errors when randomly taking endoscopic biopsies [12]. Moreover, genetic hits in tumor driver genes may lead to accelerated malignant degeneration. The majority of EAC patients are incurable because of late-stage disease at the time of diagnosis, because of a relatively late onset of symptoms. The proportion of patients with late-stage disease at the time of diagnosis has been increasing over time [2,13].

For patients with EAC undergoing treatment with curative intent, neoadjuvant chemoradiotherapy (CROSS) or perioperative chemotherapy (FLOT) regimens are the main strategies employed in the clinic. However, the majority of EAC patients poorly respond to the currently employed therapies, with a ten-year survival of 36% in the Western population according to the CROSS regimen [14]. Notably, 82% of deceased patients following the CROSS regimen are attributed to disease relapse after treatment [14]. There is an urgent clinical need to better understand specific biological mechanisms that underlie dismal outcomes and acquired resistance following treatment of EAC patients.

Targeted and immune therapies based on the unique molecular features and genomic profiles of EAC may be beneficial to EAC patients. The only targeted therapy currently applied for both locally advanced and metastatic EAC is the HER2 antibody Trastuzumab. Around 30% of EAC patients are HER2 positive and may potentially respond to this therapy [15,16]. Pembrolizumab, which blocks the interaction between PD-1 and PD-L1 is an FDA-approved immune checkpoint inhibitor for metastatic EAC. This agent is potentially beneficial to 18% of EAC patients who express PD-L1 [17]. Identifying potential molecular targets could be beneficial to the future development of personalized therapies. 

This is a comprehensive and up-to-date review, aiming to provide the latest knowledge about EAC for both researchers and clinicians. This review discusses multiple aspects of EAC, including etiology, biology, genetics and epigenetics, and clinical measures to reduce the burden of BE and EAC. Regarding the etiology, we discuss the three predominant contributors of EAC occurrence: obesity, GERD and BE. In the aspect of the biology and tumor microenvironment of EAC, we discuss driver molecules and signaling pathways associated with GERD leading to the premalignant condition BE and malignant EAC. We discuss changes in the tumor microenvironment and the consequences during the development and progression of EAC. For the review on genetics and epigenetics, we discuss the literature on distinct genetic characteristics associated with EAC among various studies. Especially for epigenetics, important findings appear in each phase of BE, LGD, HGD, early EAC and invasive EAC. Importantly, we discuss the latest measures for reducing GERD and BE, including surveillance of BE, treatment of BE, treatment of EAC and new therapies to be expected in the future.

## 2. Epidemiological and Biological Characteristics 

*Summary*: In this part, besides the epidemiological characteristics of EAC, we predominantly discuss the origin of EAC and BE, as well as the association between EAC with obesity, GERD and BE. We also discuss main driver molecules, signaling pathways and the changes in the tumor microenvironment for the development and progression of EAC. Furthermore, we discuss genetic and epigenetic characteristics by reviewing existing studies and explore possible molecular targets for personalized therapy. Why is the incidence of EAC rising despite surveillance and treatment of GERD and BE? Which driver molecules or pathways could have potential as therapeutic targets?

### 2.1. Epidemiology and Etiology

Although EAC cases only account for 14% of EC cases worldwide, it is the predominant subtype in Western countries including Western and Northern Europe, Australia, Canada and the USA [18]. In contrast to ESCC, the annual incidence of EAC has drastically increased by 767% with an average rate of 5.11% per year, from 1973 to 2017 in the USA [19]. In 2012, there were approximately 52,000 EAC cases (41,000 male and 11,000 female) worldwide [4]. In 2020, this increased to approximately 85,700, among which males are four times more often affected compared to females. Moreover, young EAC patients (< 50 years) are increasingly frequent, while there is no correlation between the age of diagnosis and survival [20]. The distribution of EAC is also affected by geographic and racial factors, suggesting a genetic predisposition. For instance, in the USA, the incidence rate of EAC from high to low displayed the following sequence: non-Hispanic whites/Hispanic whites/(American Indian/Alaska Native)/blacks/(Asian/Pacific islanders) [21]. Approximately 7% of BE and EAC cases are associated within families and presumably attributed to inherited factors [22,23]. A germline variant S631G encoded by the gene *VSIG10L* has been identified as a susceptible source of familial EAC or BE [24], which provides a lead for the detection of BE and EAC via screening in families carrying inherited factors. In addition, germline alteration in *MGST1* or *FOXP1* may be predisposing factors for BE and EAC [25,26].

Common symptoms of EAC patients are dysphagia and weight loss. A total of 79% of EAC patients present with dysphagia and 53% with weight loss, whereas 47% of patients present with a combination of both [3]. The onset of these symptoms is associated with worse prognosis of EAC [3]. GERD, obesity and smoking are risk factors for EAC [6,7,8,27,28]. The increasing incidence of EAC is likely attributed to the increasing prevalence of GERD and obesity. Studies found that abdominal obesity rather than the body mass index (BMI) predisposes to the increased risk for BE and presumably EAC [29,30,31]. GERD is most prominent in the distal esophagus [32]. Abdominal obesity can increase symptoms of GERD through increased intra-abdominal pressure [33]. Moreover, a high-fat diet underlying obesity promotes esophageal carcinogenesis in BE by inducing dysplasia due to alterations in the esophageal microenvironment and the gut microbiome [34]. Other risk factors include cigarette smoking, which approximately doubles the risk of EAC, and a low intake of vegetables and fruit [35]. Epidemiological studies showed that Helicobacter pylori infection rates are negatively associated with the incidence of EAC [36].

There is an ongoing debate regarding the cell of origin in BE even if many original studies have been performed in the past decades (Table 1). Nevertheless, it is affirmative that BE is a metaplastic condition in which the normal esophageal squamous mucosa is replaced by metaplastic columnar types of epithelia [37]. BE occurs in approximately 10% of individuals with GERD [38], whereas 40% of BE patients have no symptoms of GERD [39]. GERD not only accelerates the development of Barrett’s-like metaplasia and dysplasia [40] but also strongly contributes to the progression of BE to EAC [6]. Although 50% of EAC patients present without BE [41,42], and 40–48% of EAC patients have no or infrequent GERD symptoms [35], EAC may originate from BE even if BE is absent or unapparent at the diagnosis of EAC [43]. Studies found that EAC patients without a history of intestinal metaplasia/BE have more aggressive behavior and poorer prognosis [41,42]. 

Dysplasia is a key feature of BE, which is associated with an increased risk of progression to EAC. Progression to EAC is via cellular changes of low-grade dysplasia (LGD) and high-grade dysplasia (HGD). The presence of epithelial dysplasia remarkably affects the risk of the cancerous transformation in BE patients. The incidence rate of EAC in BE patients with no dysplasia, low-grade dysplasia and high-grade dysplasia are around 0.6%, 13.4% and 25%, respectively [10]. Other studies support that the risk rate of BE with no dysplasia on progression to EAC lies between 0.2 and 0.7% per patient per year [49,50]. The diagnosis of LGD, HGD and adenocarcinoma based on histopathological features is contentious due to low inter-observer agreement, especially for LGD [51]. Therefore, reported rates of malignant progression in patients with LGD are highly variable. One study reports a progression rate of LGD to HGD or EAC of 13.4% per patient per year [52], whereas another study reports an incidence rate of 0.84% [53]. To minimize low inter-observer agreement, AGA guidelines recommend that Barrett’s dysplasia should be confirmed by a second expert gastrointestinal pathologist [54]. Definitive and simplified histopathological criteria and larger biopsy specimens to avoid sampling bias are needed to improve inter-observer agreement. Barrett’s segment length is another risk factor for neoplastic progression in BE. Barrett’s segment length is significantly associated with the risk of malignant progression of non-dysplastic Barrett’s esophagus (NDBE) [55]. One study found that the risk of malignant progression increases by 19% per centimeter of Barrett’s segment length [56]. One study found that the incidence rates of EAC progressed from NDBE with short-segment BE (< 3 cm) and long-segment BE (≥ 3 cm) are 0.24% and 0.76%, respectively [57]. 

### 2.2. Histology

According to WHO classification, EAC is classified into three subgroups based on the percentage of gland formation: poorly differentiated (<50%), moderately differentiated (50–95%) and well differentiated (≥95%) [58]. This classification shows no correlation with survival [59]. Moreover, EAC can be classified according to Lauren’s classification which distinguishes the intestinal type, diffuse type and mixed type of EAC. One study showed that Lauren’s classification subtypes can successfully predict response to chemotherapy and survival [60]. EAC classified as diffuse and mixed types are associated with less pathological response to chemoradiotherapy and poor prognosis [61]. A more analytical approach to identifying histological subtypes that provide useful prognostic information is an ongoing topic of research. One study classified EAC cases into histological subtypes including mucinous muconodular carcinomas, invasive mucinous carcinomas, diffuse desmoplastic carcinomas, diffuse anaplastic carcinomas and mixed carcinomas, which showed prognostic significance [59]. Also, another histologic classification of EAC into papillary, tubular, mucinous and signet ring subtypes has been proposed, but this classification lacks evidence of clinical impact [62]. Subgrouping based on other features than histology is discussed later on in “Genetics” and “Epigenetics”. 

### 2.3. Biology and Immunology

EAC initiation is driven by GERD through reprogramming of cell proliferation and differentiation in the esophageal mucosa [63]. Long-standing GERD causes metaplasia of the esophageal squamous epithelium in a subset of patients through dysregulation of multiple driver molecules and pathways (Figure 1). GERD activates the NF-κB signaling in esophageal squamous cells [64,65,66], a pathway that is associated with abnormal cell proliferation and differentiation, treatment resistance and metastasis in multiple cancers [67]. In EAC patients, the expression of NF-κB is negatively correlated with complete pathologic response to neoadjuvant chemoradiotherapy [68]. GERD also increases the expression of CDX2 [69], which drives the development of an intestinal-type metaplasia [70,71,72]. One study identified activation of CDX2 in the development of BE [43]. GERD activates the *CDX2* promotor through the activation of NF-κB and upregulates the expression of the CDX2 protein, leading to the production of the intestinal-type protein MUC2 [73]. GERD induces the expression of Notch ligand Delta-like1 (Dll1) via a CDX2-dependent pathway in the development of BE, and Dll1 expression is significantly higher in BE than in normal human esophageal squamous epithelium [74]. Dll1 expression facilitates intestinal-type metaplasia in esophageal squamous cells in combination with CDX2 expression. BMPs, especially BMP4, are highly expressed in BE and EAC, and BMP4 alone or in combination with CDX2 drives the differentiation of columnar epithelia [75,76]. BMP2 and BMP4 are both highly expressed in EAC, which may enhance the aggressiveness of EAC through triggering of non-canonical BMP signaling [77]. Moreover, COX-2 regulated by NF-κB exerts an important role in carcinogenesis in multiple cancers [78]. The expression of the COX-2 protein is significantly higher in patients with BE and EAC compared to normal squamous esophageal epithelia in healthy patients [79,80]. Genetic variants of the *COX-2* gene are significantly associated with an increased risk of EAC [81,82]. Targeting COX-2 has been suggested as a potential therapy in EAC. Inhibition of COX-2 with the inhibitor MF-Tricyclic reduces the incidence of EAC in a mice model of BE [83]. Interestingly, one study reported that epithelial-to-mesenchymal transition (EMT) is associated with the progression from early EAC to invasive EAC. The measurement of EMT markers prior to intervention may help to make the clinical decision between endoscopic submucosal dissection or esophagectomy as the preferred treatment [84].

Cytokines and chemokines are important components of the tumor microenvironment. Cytokines mediate the inflammatory context in the tumor microenvironment, whereas chemokines are directly associated with the malignant behavior of tumor cells, for instance, cell migration. The change in cytokines and chemokines caused by GERD in the esophageal epithelium contributes to the development and progression of BE and EAC. GERD causes overexpression of the pro-inflammatory cytokines IL-1β, IL-8 and IL-6 in esophageal epithelium cells [85]. IL-1β is overexpressed in both BE and EAC compared to normal squamous esophagus [32,86]. IL-1β overexpression in the mouse esophagus induces overexpression of BMP4, SHH and CDX2, and the gene expression profile of the IL-1β-overexpressed mouse esophagus closely resembles that of human BE and EAC [40]. This suggests that IL-1β induces the development and progression of BE and EAC. In addition, IL-8 expression is significantly higher in patients with BE or EAC than in those who have only GERD symptoms [87]. The maximal degree of inflammation with a significant increase in pro-inflammatory IL-1β and IL-8 expression is located at the new squamo-columnar junction, whereas the minimal degree of inflammation with a significant increase in the anti-inflammatory IL-10 expression is located at the distal portion of the BE segment where most EAC occur [32]. IL-10 expression in EAC is associated with worse overall survival [88]. One study reports that IL-6 produced by cancer-associated fibroblasts is significantly higher in EAC than in normal tissue, which leads to resistance of EAC cells to chemoradiotherapy (CROSS regimen) [89]. The anti-inflammatory microenvironment existing in EAC and its peritumoral tissue is negatively associated with the survival of patients [88]. Moreover, one study found that the expression of CXCR4, a chemokine receptor, increases with the progression from BE to LGD and HGD and EAC in a mouse model [85]. CXCR4 and its chemokine ligand CXCL12 are highly expressed in EAC, and the expression is associated with poor prognosis and lymph node metastases [85,90]. High expression of the chemokine receptor CXCR7 and its chemokine ligand CXCL12 in EAC is also associated with poor prognosis [91]. In addition, the chemokines, chemokine receptors, cytokines and interleukins CXCR1, CXCR2, CXCL1, CXCL2, CXCL3, CXCL6, CXCL8, CCL15, CCR4, IL-2, IL-6, IL-7, IL-8, IL-15 and IL-18 are significantly higher expressed in HGD/EAC compared to LGD/BE [92]. 

The immune microenvironment plays a key role in the development and progression of BE and EAC. The immunosuppressive Foxp3 *FOXP3* is significantly higher expressed at the mRNA level in BE than in normal squamous epithelium [93]. The number of FOXP3^+^ lymphocytes is significantly higher in BE and EAC than in non-metaplastic esophagitis [94] and is associated with high Ki-67 expression [95]. Ki-67 is higher expressed in EAC than in BE and higher in BE than in healthy person [96]. In addition, the number of FOXP3^+^ lymphocytes is significantly higher in HGD than in LGD and higher in LGD than in NDBE [95]. The number of CD3^+^ lymphocytes is also higher in BE tissues adjacent to EAC than in BE tissues without EAC and is associated with the absence of Barrett’s metaplasia [95]. Another study also found that the number of *CD3*^+^ FOXP3^+^ regulatory T cells significantly increases from BE to LGD and to EAC [92]. A high CD8^+^ lymphocyte number in BE adjacent to EAC is associated with worse overall survival and lymph node metastasis [95]. CD8^+^ cytotoxic T cells and CD163^+^ macrophages are both decreased in EAC compared to HGD [92]. BE is associated with a transition from a pro-inflammatory Th1-type immune response to an anti-inflammatory Th2-type immune response [97]. The number of Th2 effector cells (plasma cells and mast cells) is higher in BE than in reflux esophagitis, whereas the number of Th1 effector cells (macrophages and CD8^+^ T cells) is significantly lower in BE than in reflux esophagitis [98]. Another study also confirmed that BE harbors a Th2-predominant cytokine profile compared to the pro-inflammatory nature of esophagitis [99]. In addition, CTLA4 expressing in regulatory T cells with an anti-inflammatory role in EAC is upregulated, which is associated with poorer overall survival [88]. M1 macrophages are associated with tumor inhibition through intrinsic phagocytosis and increased anti-tumor inflammatory responses, whereas M2 macrophages are assumed to be tumor-promoting by involvement in stromal activation and remodeling, angiogenesis, neovascularization and immuno-suppression [100]. The infiltration of CD8^+^ T cells and M1 macrophages is significantly lower in EAC compared to the adjacent normal esophageal tissue [101]. M2 macrophages are significantly increased in EAC compared to HGD [92]. Eosinophils are significantly decreased in EAC compared to BE, LGD or HGD [92]. Eosinophilic infiltration in cancer has been associated with better prognosis [102]. Compared to the adjacent normal esophageal tissue, the microenvironment of EAC is characterized by infiltration with T regulatory cells and effector T cells, expansion of plasmacytoid dendritic cells, increased expression of cancer-associated fibroblasts and reduction in endothelial cells. Interestingly, a reversal of these characteristics happens following treatment with the FLOT regimen [103]. Another study shows that poor infiltration of cytotoxic effector cells and increased immune inhibitory signaling are the main characteristics of the EAC microenvironment [92].

### 2.4. Genetics

The esophageal mucosa is recurrently exposed to gastric and bile acid refluxates caused by GERD. This may cause DNA damage and the formation of mutational patterns, such as an A > C transversion [104,105]. A common mutational pattern T: A > G : C also exists in EAC [105,106]. This mutational signature is similarly enriched in SNVs of both BE and EAC, suggesting a common pathogenic etiology [107], which might be the result of long-standing gastroesophageal reflux [105,106,108].

High inter- and intra-tumor heterogeneity is a significant feature of EAC. Extensive genomic heterogeneity commonly presents in the advanced stage of cancer [109]. EAC is characterized by a high mutational burden, chromosomal instability (CIN), copy number variations (CNVs) and highly variable mutational signatures. The mutational rate in EAC is more significant compared to other types of cancer including gastric cancer, pancreatic cancer, colorectal cancer and hepatocellular carcinoma [110]. EAC harbors a median of 26,161 genome-wide mutations per tumor with a median mutational frequency of 9.9 mutations/Mb (range of 7.1–25.2/Mb) [105]. In comparison, 2.64 mutations/Mb (range 0.65–28.2/Mb) are seen in pancreatic cancer and 5.801 mutations/Mb (range 0.725–88.470/Mb) in gastric cancer [111,112]. One study showed a significant correlation between mutation frequency and the pathological grade of dysplasia (NDBE vs. DBEs) [113]. However, another study found that the mutation rate poorly correlates with dysplasia grade [107].

Genetic clonal diversity is a significant feature of EAC [114]. The prevailing view in EAC is that a gradual accumulation of mutations drives the transformation of precancerous lesions to EAC. Inactivation of the tumor suppressor *CDKN2A*, and clonal or polyclonal expansion in NDBE, followed by a dysplastic clone with *TP53* inactivation and other somatic genetic variations, is one possible sequence of EAC development [11,115]. However, another study demonstrated that EAC initiation is through a more direct way in which *TP53*-mutant cells undergo genome doubling, followed by the acquisition of oncogenic amplifications rather than through the gradual accumulation of tumor-suppressor alterations [113]. It is known that carcinogenic factors drive abnormal clones that erode and occupy most regions of esophageal mucosa through clonal expansion. Genomic analyses of the normal esophageal epithelium (upper- and mid-esophagus) showed that strong expansion of clones with cancer-associated mutations in *Notch1* and *TP53* occurs in up to 80% and 37% of healthy middle-aged and elderly people, respectively [116]. Notably, the mutational burden of *Notch1* is several times higher in the normal esophagus than in EC [116]. This indicates a complex somatic clonal evolution within normal esophageal tissue. Measures of genetic clonal diversity have been identified as robust biomarkers to stratify for cancer risk in endoscopically surveyed BE patients [117].

Notably, point mutations are reported to be dominant in EAC [105]. Some susceptibility genetic loci involved in the embryonic development of the esophagus, for instance, *FOXF*, *BARX1* and *ABCC5*, were identified to be associated with BE and EAC risk [118,119]. To date, many mutated genes of EAC have been identified from EAC genome sequencing studies. We identified seven studies considering the analysis of mutated genes in EAC. The most frequently reported significantly mutated genes in EAC by DNA sequencing include *TP53*, *SMAD4* and *CDKN2A*, which are reported in all seven EAC studies [1,105,106,120,121,122,123]. The rest mutated genes are reported in one or few studies (Table 2). The discrepancy in mutated genes identified from studies is likely due to different detecting methods, small study cohorts (between 112 and 551 patients) and distinct demographics. Targeted sequencing or exome sequencing as used in the three studies mentioned above [1,105,122] is not able to detect large-scale structural rearrangements and heterogeneity that exist in EAC [108]. In contrast, whole-genome sequencing (WGS) as applied in four studies [106,120,121,123] is able to more comprehensively profile genomic alterations in EAC. 

*TP53* is a tumor suppressor gene and is widely mutated in a variety of human cancers including EAC [124]. One study found that the majority of mutated genes in EAC are also mutated at the NDBE stage, with the exception of *TP53*, which is only mutated in HGD (72%) and EAC (69%), and *SMAD4*, which is only mutated in the EAC stage (13%) [106]. *TP53* mutation cannot differentiate between high-grade dysplasia and EAC. However, in EAC, *TP53* mutations are significantly correlated with the grade of histologic differentiation and worse survival [125]. *SMAD4* is the only identified gene mutation that is able to differentiate between EAC and precancerous stages. However, *SMAD4* mutations are only found in about 13% of EAC [106]. SMAD4 loss in EAC is also correlated with disease recurrence and poor survival [77,126]. 

One study reported that chromosomal structural rearrangements and copy number variations drive the development of EAC [123]. The genes with CNVs in EAC are reported in multiple studies (Table 3). The genomes of EAC are highly rearranged and complex (median of 263 structural variants (SVs) per tumor, range 126–776) [108]. As a pattern of chromosomal structural rearrangements, chromothripsis has been reported to present in 32.5% of EAC [108], whereas the prevalence of chromothripsis in other cancers is only 5% [127]. One study reports that the number of chromosomal aberrations is significantly correlated to patient survival in EAC [128]. 

In contrast to the gradual accumulation of somatic mutations (short deletions/insertions or single-nucleotide substitutions), chromosomal structural rearrangements occur relatively late in cancer progression and may occur within a short time [131,132]. Copy number variations are significantly increased in EAC compared to BE [107]. The levels of genomic instability are positively correlated with advanced stages of EAC [129]. A longitudinal study of BE to EAC progression found that progressors acquire significantly more somatic chromosomal alterations and genomic diversity during a median period of four years before EAC diagnosis, whereas the genomes of non-progressors remain stable over prolonged periods of time [11]. The frequency of copy number variations has been found to be low in the case of no dysplasia (1.3%) and increases along with the stages of dysplasia (4.7%) and reach high levels in EAC (30%) [130]. In EAC, copy number gains occur mostly for chromosomes 7, 8, 19 and 20, whereas chromosomes 5, 9, 18 and 21 are more common for copy number losses [130]. Several structural genetic alterations in specific genes, including copy number variation and loss of heterozygosity, are associated with the progression of BE to EAC. The loss of heterozygosity of the *TP53* gene at chromosome 17p (17p (p53) LOH) is a predictor of progression to EAC [133]. Moreover, analysis of the abnormalities of the genes *CDKN2A* and *MYC* and aneusomy significantly improved risk prediction for the progression of BE to EAC [134]. Furthermore, a combination of 9p LOH, 17p LOH and DNA content tetraploidy and aneuploidy proved to be significant, independent predictors for EAC risk [135]. Moreover, amplification of potential therapeutic targeted genes, for instance, receptor tyrosine kinases (RTKs) *HER2*, *EGFR*, *MET* and *FGFR*, widely exists in EAC [105,123,136]. For instance, high-level amplifications of *HER2* and *EGFR* are found in 17% and 11% of EAC, respectively [123]. *HER2* amplification and overexpression are associated with poor survival of EAC patients [137]. Co-amplification of different RTKs, for instance, *HER2* and *EGFR*, is frequent in EAC and may cause resistance to anti-RTK therapies [123]. Therefore, a combination of RTK antibodies/inhibitors to circumvent tumor resistance may be an effective therapeutic strategy. Interestingly, several studies found that shorter telomere length in BE patients is associated with an increased risk of progression to EAC [138,139,140]. Obesity and cigarette smoking can reduce telomere length [141,142]. In addition, telomere integrity analysis based on whole-genome sequencing found somatic telomere shortening in EAC, which is associated with complex chromosomal rearrangements [108]. This means that telomere length could be applicable to the risk stratification of Barrett patients.

Subclassifications are of paramount importance for the development of new targeted therapies and enable identification of new targets in a specific subset of patients. Nones et al. categorize EAC into three subtypes using structural rearrangement patterns: unstable genomes (tumors with ≥450 structural variants (SVs)), scattered (<450 SVs evenly distributed across the genome) and complex localized (with a concentration of SVs in a single or few chromosomes) [108]. The association between these subtypes and prognosis or therapy response needs to be investigated in the future. Based on mutational profiles, EAC can be divided into three subtypes: (i) enriched for BRCA signature in the homologous recombination pathway; (ii) a dominant T > G mutational pattern; and (iii) a C > A / T mutational pattern. These groups provide a strategy for therapy selection [123].

Next to genomic abnormalities, transcriptomic profiling by microarray or RNA sequencing has been an important topic of research [143]. Based on RNA profiling, the TCGA research group investigated gastric adenocarcinoma and proposed a new classification with four subtypes: chromosomal instability (CIN), microsatellite instability (MSI), genomic stability (GS) and Epstein–Barr-virus-positive cancers [144]. In an additional study, the TCGA group showed that EAC has a high frequency of chromosomal instability (CIN) and therefore resemble one of the four subtypes of gastric cancer. In this study, 71 out of 72 EAC cases were identified to have CIN indicating similarities between EAC and gastric cancers with CIN [1]. More recently, our research group was able to further differentiate EAC into three molecular subgroups, characterized by (i) p38 MAPK/Toll-like receptor signaling, (ii) an activated immune system and (iii) impaired cell adhesion, and this classification was associated with response to neo-adjuvant treatment [143]. 

Although much effort has been made to understand the mechanism underlying genetic differences, only few critical mutations and chromosomal structural rearrangements have been identified from the comparison of pre-EAC lesions and EAC. It is challenging to identify molecular biomarkers for predicting the progression of precursor lesions to EAC. These biomarkers are critical, for instance, as tools for the identification of high-risk populations for population screening programs. At present, next to RNA biomarkers, the analysis of epigenetics may hold promise to identify EAC subtypes and provide therapeutic targets for EAC. 

### 2.5. Epigenetics

The role of DNA methylation has been studied in BE and the development of EAC (Table 4). The levels of DNA methylation in BE and EAC are similar and significantly higher than in normal esophageal tissue [145]. Genes that are frequently methylated in many cancers including *APC*, *ID4*, *MGMT*, *SFRP1*, *TIMP3* and *TMEFF2* have similar methylation frequencies in EAC and BE, whereas *CDKN2A* and *RUNX3* are significantly more frequently methylated in EAC than in BE [146]. The hypermethylation of the *CDKN2A* promoter is associated with the progression of BE to EAC [147,148]. Moreover, in many EAC, the *MT3* gene is hypermethylated, which is associated with advanced tumor stages [149]. One study reported that BE progressing to EAC harbors widespread DNA methylation that particularly occurs in 70% of known imprinted genes, compared to BE cases that do not progress to EAC [150]. Based on the number of methylations of four genes, respectively, *SLC22A18*, *RIN2*, *PIGR* and *GJA12*, stratification of BE patients into three risk groups (low, intermediate and high) with prognostic significance could be performed [150].

MicroRNAs (miRNAs) have been identified as oncogenes or tumor suppressors through their roles in regulating the expression of target genes in EAC. Altered expression of microRNAs can affect the development and progression of EAC (Table 5). One study identified 13 differentially expressed miRNAs during the development of BE (when comparing BE with normal squamous esophageal epithelium). These genes include upregulated miR-215, miR-560, miR-615, miR-192, miR-326 and miR-147 and downregulated miR-100, miR-23, miR-605, miR-99, miR-205, let-7c and miR-203 [151]. MiR-101, miR-125, miR-197, miR-200 and miR-513 are upregulated, and miR-20, miR-23, miR-181, miR-193, miR-203 and miR-636 are downregulated in the progression from BE with LGD to BE with HGD [152]. MiR-28, miR-30 miR-126, miR-143, miR-145, miR-181 and miR-199 are upregulated, and Let-7, miR-193, miR-345 and miR-494 are downregulated in the progression from HGD to EAC [152]. MiR-25, miR-93 and miR-106b are overexpressed in EAC compared to BE [153]. MiR-200a is overexpressed in EAC compared to the normal esophagus. Its expression decreases with advanced stages of EAC, which suggests that miR-200a may be involved in the early phase of the development of EAC [154]. MiR-133b is downregulated in EAC compared to the normal esophagus [155]. Besides differential expression, targeted genes and pathways of a subset of miRNAs were investigated. MiR-133b may regulate the proliferation and apoptosis of EAC cells by targeting the pro-survival gene *MCL-1* and the receptor tyrosine kinase MET signaling pathway [156,157]. Moreover, miR-21 is overexpressed in EAC and is thought to play a role in the initiation and development of EAC by disrupting apoptosis due to activation of Ras/MEK/ERK signaling and NF-κB signaling [155,158,159,160]. Our group studied several miRNAs in EAC and demonstrated that miR-125a reduced MHC-I expression, which is required for antigen presentation and T-cell response. Reduced MHC-1 expression and adaptive immune system markers were associated with improved patient outcomes in EAC [161].

## 3. Management of EAC

Summary: Effective clinical management of BE and EAC is essential to reduce EAC incidence and improve the quality of life and prognosis of EAC patients. In this part, we discuss the latest clinical advancements for the prevention and treatment of EAC. The implementation of CROSS and FLOT chemo(radio)therapy regimens brought benefits to the potentially resectable EAC patients, whereas almost no progress in the treatment for patients with distant metastatic EAC has been made. What studies should be performed to improve the treatment of EAC patients with distant metastasis?

### 3.1. Prevention of EAC in Patients with BE

Decreasing risk factors by maintaining an appropriate BMI, frequent intake of vegetables and fruit and prohibition of cigarette smoking are general lifestyle measures for the prevention of EAC. Endoscopy as a population screening method to detect patients with BE has not been recommended yet considering the relatively high costs and risks of complications [162]. This means that many cases of BE are not diagnosed, especially because a large part of the BE population never have GERD symptoms or have only mild symptoms until the advent of EAC. There is an unmet clinical need for screening tools in the general population to detect BE. Low-cost and low-risk methods such as non-endoscopic screening by sponge capsules and breath tests are under investigation [163,164]. 

Early diagnosis and successive monitoring of BE through endoscopic surveillance do play an effective role in the prevention of EAC. Surveillance programs are effective, and the incidence of invasive EAC is lower, while outcomes of EAC patients detected in BE surveillance cohorts are better [165]. Patients with BE undergoing endoscopic surveillance can be treated in the early curative stages of EAC and have significant survival benefits compared to EAC patients who have not been detected in endoscopic surveillance programs [166,167,168]. Besides endoscopic surveillance, chemo-preventive medicines can effectively reduce progression to EAC in BE patients. The use of proton pump inhibitors (PPIs) effectively reduced the risk of EAC progression in patients with BE [169,170,171]. Aspirin and other nonsteroidal anti-inflammatory drugs (NSAIDs) have been demonstrated to reduce the risk of EAC compared to controls [135,172,173,174,175]. Aspirin and NSAIDs both inhibit the generation of cyclooxygenase (COX) [176], and aspirin also inhibits the activity of NF-κB signaling [177]. 

### 3.2. Treatment of EAC

Patients with BE without dysplasia are included in long-term endoscopic surveillance programs and are treated with PPIs. BE patients with dysplasia may undergo endoscopic therapies, including photodynamic therapy, cryotherapy, radiofrequency ablation (RFA) and endoscopic mucosal resection (EMR). EMR combined with RFA is the main therapeutic strategy currently applied. One study showed that photodynamic therapy is highly effective for eradicating dysplastic BE, but it is associated with a relatively high rate of complications [178]. Cryotherapy displayed effective results as a treatment for BE [179] and enables eradication of intestinal metaplasia in 55% of patients, dysplasia in 85–90% and HGD in 95–100% [180]. RFA achieves complete eradication of dysplasia in 81–91% of patients and is characterized by low recurrence rates [181]. BE patients with early-stage cancers (tumor in situ, staged as T1a) either undergo endoscopic treatment or surgery [182].

Patients with EAC undergo tumor staging to select the most appropriate treatment regimen. Endoscopy with endoscopic ultrasound and computed tomography (CT) or positron emission tomography (PET) with [^18^F]2-fluoro-2-deoxy-d-glucose (FDG) are frequently used in the clinic [183]. Early, locally advanced EAC without distant lymph node involvement and distant metastasis can be amenable to treatment with curative intent. Neoadjuvant chemoradiotherapy followed by surgery (CROSS regimen) is applied in the clinic. Neoadjuvant chemoradiotherapy (nCRT) with the administration of carboplatin and paclitaxel for 5 weeks and concurrent radiotherapy (41.4 Gy in 23 fractions, 5 days per week) lead to a significant survival benefit compared to surgery alone [184]. Another successful and clinically applied perioperative chemoregimen is FLOT (fluorouracil plus leucovorin, oxaliplatin and docetaxel), which improves overall survival compared to perioperative ECF/ECX (epirubicin and cisplatin plus either fluorouracil or capecitabine) [185,186]. A comparison of CROSS and FLOT has also been investigated in clinical trials, with one study showing comparable survival benefits [187], while other studies are ongoing [188].

Regarding the surgical approach for EAC, radical transthoracic esophagectomy in combination with lymphadenectomy is the first choice of treatment [183]. In addition, minimally invasive esophagectomy (MIO), which is characterized by a lower post-operative morbidity, better quality of life and quicker functional recovery compared to open esophagectomy, has been introduced to clinical practice recently [189,190].

Definitive CRT is an option for those EAC patients who are not operable due to age or co-morbidities or are unwilling to undergo surgery [191]. For these therapies, adjusted CROSS and FOLFOX regimens or a regimen of fluorouracil plus cisplatin are frequently used [192].

For EAC patients with stage IV disease due to distant metastasis, palliative treatment with systemic therapy is currently the treatment of choice. This treatment includes fluoropyrimidine (fluorouracil or capecitabine) combined with either oxaliplatin or cisplatin and is a regimen according to the NCCN guidelines [193,194]. The FDA also approved Lonsurf (trifluridine/tipiracil) for the treatment of metastatic EAC. This is based on a phase III trial that showed that trifluridine/tipiracil significantly improves overall survival in metastatic EAC compared to a placebo group [195]. Platinum-fluoropyrimidine doublet (oxaliplatin and cisplatin plus fluoropyrimidines) can also be considered as a treatment for advanced and metastatic EAC [196]. 

As for now, only few targeted and immune therapies have been approved for the treatment of EAC (Table 6). HER2, VEGFR2 and PD-1 are the only three molecular targets. Given that currently employed targeted and immune therapies marginally improved the outcomes of patients, exploring more available molecular targets is still an imperative clinical need. 

## 4. Discussion

The majority of treatments for EAC currently employed in the clinic are not specifically designed for EAC based on its biological features. Although the CROSS and FLOT regimens displayed better outcomes than other regimens, their establishment is derived from clinical trials and is less related to the unique molecular features of EAC. Chemoradiotherapy is a common approach for the treatment of most cancer types. The inherent non-specificity of chemoradiotherapy can cause severe side effects and discontinuation of the treatment. Also, it is thought that chemoresistance induced by chemoradiotherapy may accelerate the metastatic behavior of residual cancer cells. Nevertheless, the currently employed targeted and immune therapies play only an auxiliary role in the treatment of EAC. The therapeutics Trastuzumab (anti-HER2), Ramucirumab (anti-VEGFR2), Nivolumab (anti-PD1) and Pembrolizumab (anti-PD1), which are used in the clinic for EAC, are monoclonal antibodies, whose frequent side effects have been increasingly drawing the attention of researchers and clinicians and whose optimal alternatives are under investigation [203]. The potential therapeutic targets, for instance, the receptor tyrosine kinases EGFR, MET and FGFR in EAC, still hold a challenge due to their failed clinical trials [33]. For the prevention and treatment of EAC, there are certainly other methods. At present, endoscopy is the only surveillance method for BE in the clinic. Unfortunately, it has not been recommended for screening people with GERD or at risk for EAC as it is costly and invasive. Therefore, a subset of BE patients who develop EAC are missed during the optional phase of progression for effective endoscopic treatment. There is still a big gap between fundamental research and the ideal therapy for EAC. Therefore, the development of more effective therapies is an unmet medical need. 

This work systematically summarized and extracted valuable findings from a large number of existing studies. As such, we hope to provide a framework for future research. This work might accelerate key research and the design of novel translational studies. On the molecular level, we discussed the most important factor associated with BE and EAC, which is GERD. GERD is known to erode and damage esophageal mucosa and induce BE. It also acts on the tumor microenvironment in EAC. However, the underlying comprehensive biological mechanisms of GERD on BE and EAC are less reported. Previous studies vaguely describe premalignant promotion as the main biological mechanism of GERD driving EAC without mentioning specific molecules and pathways [204]. We summarized that GERD acts by impacting both esophageal squamous epithelial cells and the tumor microenvironment, which includes important molecular and signaling pathways, such as CDX2, COX2, BMP4, MUC2, Dll1 and the NFkB pathway. In current clinical practice, chemopreventive medicines (Aspirin and NSAIDs) against COX2 and NFkB have been used for the preventive treatment of GERD and BE [176,177]. This review provides the rationale for future translational studies to focus on targeting CDX2, BMP4, MUC2 and Dll1; especially CDX2 and BMP4 are targets of interest, considering that MUC2 and Dll1 are inhibited in the case of inhibition of CDX2. Moreover, many studies have focused on the change and role of cytokines and chemokines in the tumor microenvironment of EAC. We found that the biological interaction between these factors in the tumor microenvironment is a complex network. For instance, IL-1β mainly secreted by M2 macrophages might upregulate BMP4 and CDX2 expression in esophageal epithelial cells and SHH expression in cancer-associated fibroblasts. BMP4, CDX2 and SHH are both involved in the pathogenesis of BE and EAC [77,205,206]. Therefore, targeting IL-1β or IL-1β-secreting cells is an interesting potential treatment strategy. Previous studies showed that EAC is characterized by an anti-inflammatory and immunosuppressive microenvironment with potential therapeutic targets within this microenvironment. FOXP3, as an immunosuppressive protein expressed in BE cells, promotes dysplasia and cell proliferation. FOXP3 is expressed in lymphocytes, especially in regulatory T cells, and its expression inhibits the immune system and reduces the destruction of cancer cells. Fortunately, an anti-FOXP3 inhibitor has been tested in a Phase I clinical trial, which showed promising anti-tumor effects [207]. Therefore, inhibition of FOXP3 could be an attractive approach to prevent and treat EAC. Tumor-associated macrophages are an essential determinant of the development and progression of tumors and display anti-tumor potential in other types of cancer [208]. In this review, we found that M1 macrophages act as Th1 effector cells and have tumor-inhibiting effects in the tumor microenvironment in EAC. Macrophage-centered therapy has been investigated in several types of cancer but not in EAC [209]. In the future, studies should focus on M1-promoting and M2-inhibiting polarization in the tumor microenvironment of EAC. Specifically, it should be determined which cytokines and chemokines are involved in the polarization of M1 or M2 macrophages to enable the development of corresponding inhibitors or activators.

There is a controversy about whether the cell of origin of BE originates from gland ducts or from the gastric cardia. The resolution of this controversy will be a milestone in better understanding the pathogenesis of BE and EAC in the future. The difficulty in the confirmation of the origin of BE is mainly attributed to the inability to observe metaplastic transformations in vivo due to a lack of reliable animal models [210]. Our group developed a mouse model resembling BE that regenerates the neo-columnar epithelium [211], which may accelerate the progression of this kind of study.

By summarizing alterations on the genetic and epigenetic level in EAC, we reviewed the potential clinically applicable biomarkers to complement the insufficiency of histopathological grading of EAC. However, this is a difficulty for EAC considering that the genetics of EAC is characterized by a high mutational burden and a variety of mutational signatures, complemented by a high level of chromosomal instability. Also, the epigenetics of EAC is featured by massive microRNA variations and various DNA hypermethylations. In light of this, the purpose of precision treatment through carrying out more accurate gene stratification might be beneficial for the treatment of EAC. In different studies, it has been demonstrated that EAC can be divided into various molecular subgroups. One subclassification of EAC can be used to predict response to neoadjuvant treatment for EAC patients [143]. However, the subclassifications in several studies seem to have less clinical significance [108,123]. The definition of the molecular subtypes of EAC will be a milestone in the future. By reviewing the genetic and epigenetic studies, *TP53*, *SMAD4* and *CDKN2A* are concluded to be biomarkers with more potential for the prediction of prognosis and treatment of EAC than other mutated genes as they are the most reported mutated genes. Studies reported that *TP53* and *SMAD4* are both prognostic factors, which can predict survival time or response to neoadjuvant therapy in EAC [126,212], whereas no study has reported *CDKN2A* as a prognostic factor in EAC. Other mutated genes discovered by one or few studies need to be further investigated. Besides detection methods and cohort size, mixtures with non-tumor cells (for instance, stromal cells and immune cells) in biopsies or resected specimens is an important factor that causes inconsistent findings between sequencing or gene expression data. Inconsistent gene profiling data could impact the identification of subgroups and the assignment of more suitable tailor-made therapies. This should be handled, for instance, by cell sorting or deconvolution methods including the implementation of single-cell sequencing. Epigenetic changes vary with each distinct phase of BE, LGD, HGD, early EAC and late EAC. New biomarkers that can be developed based on specific epigenetic changes will be suitable for patients in the specific phase of the disease. We found that some studies reported on genomic abnormalities as observed in single samples taken at one location and at one time point and neglected the temporal and spatial differences that are present in BE and EAC. Cancer studies of slow progressive precancerous lesions require longitudinal studies considering sufficient sample size, reasonable negative control populations and tissue sampling with follow-up that is synchronized with disease progression. Such studies may be more effective in revealing potential druggable targets compared to cross-sectional studies which are prone to bias due to patient heterogeneity.

## 5. Conclusions

In this work, we revealed up-to-date knowledge of biological characteristics and treatment strategies of esophageal adenocarcinoma, including molecular pathogenesis, tumor microenvironment, genetics, epigenetics and therapy advancement. Importantly, we discussed the questions that arose from these studies and the clinical practice and the scientific milestones, which may be useful to improve EAC prevention and treatment in the future.

## Figures and Tables

**Figure 1 cancers-15-05410-f001:**
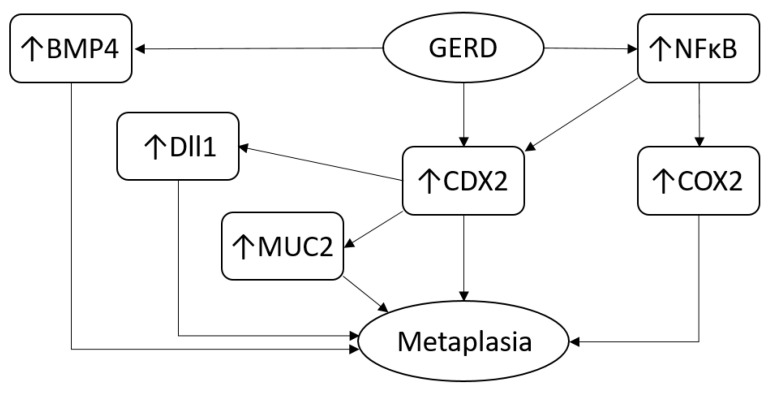
Schematic overview of important molecular pathways driving metaplasia in esophageal squamous epithelium as a result of GERD.

**Table 1 cancers-15-05410-t001:** The existent theories about the origin of Barrett’s esophagus as suggested by original research.

Reference	Authors	Year	Origin	Drivers
[43]	K. Nowicki-osuch et al.	2021	Gastric cardia	c-MYC- and HNF4A-driven transcriptional programs
[44]	D. Straub et al.	2019	Non-squamous cells residing in submucosal gland ducts	Glycine-conjugated bile acids
[45]	C. Minacapelli et al.	2018	Normal esophageal squamous epithelial cells	Acid and bile
[46]	M. Jiang	2017	p63^+^KRT5^+^KRT7^+^ basal cells in the upper gastrointestinal tract	Ectopic expression of CDX2
[40]	M. Quante	2012	Gastric cardia stem cells	Bile acids and/or nitrosamines
[47]	X. Wang et al.	2011	Embryonic cells at the squamocolumnar junction	Competitive interactions between cell lineages
[48]	S. Leedham	2008	Squamous gland ducts situated throughout the esophagus	Gene mutations

**Table 2 cancers-15-05410-t002:** Mutated genes in EAC as suggested by original research.

Reference	Authors	Year	Mutated Genes	Detecting Methods
[122]	A. Orsini et al.	2023	*TP53*, *ATM*, *MSH6*, *APC*, *PIK3CA*, *SMAD4*, *CDKN2A*, *SMARCA4*, *ERBB2*, *HNF1A*, *CHEK2*, *FLT3*, *PTEN*, *IDH2*, *CTNNB1*, *MET*, *STK11*, *ALK*, *KRAS*, *RET*, *EGFR*, *ARID2*, *CDK6*, *TSSC1*, *MAP2K1*, *SRC*	Targeted sequencing
[120]	A. Frankell et al.	2019	*TP53*, *CDKN2A*, *KRAS*, *MYC*, *ERBB2*, *GATA4*, *CCND1*, *GATA6*, *SMAD4*, *CDK6*, *ARID1A*, *EGFR*, *CCNE1*, *CCND3*, *MUC6*, *MDM2*, *KCNQ3*, *APC*, *SMARCA4*, *PIK3CA*, *ABCB1*, *PTEN*, *MET*, *RNF43*, *DNAH7*, *TSHZ3*, *LRRK2*, *TRPA1*, *NAV3*, *ARID2*, *SLIT2*, *EPHA3*, *SCN3A*, *CRISPLD1*, *AXIN1*, *FBXW7*, *PPM1D*, *ACVR2A*, *RASA1*, *CD1A*, *CCDC102B*, *CHL1*, *LIN7A*, *COIL*, *MAP2K7*, *EPHA2*, *PBRM1*, *POLQ*, *ARID1B*, *CTNNB1*, *SIN3A*, *RPL22*, *PIK3R1*, *MAP3K1*, *NIPBL*, *B2M*, *FAM196B*, *HIST1H3B*, *TGFBR2*, *MBD6*, *BRAF*, *MSH3*, *CHD4*, *CDH1*, *GATAD1*, *KDM6A*, *CDKN1B*, *ACVR1B*, *STK11*, *NOTCH1*, *ZFHX3*, *JAK1*, *PCDH17*, *ELF3*, *GPATCH8*, *C3orf62*	Whole-genome sequencing
[121]	D. Lin et al.	2018	*PIK3CA*, *PBRM1*, *SMARCA4*, *CTNNB1*, *PCDH18*, *C6orf114*, *CHRNB1*, *EPHA2*, *SEMA5A*, *PGCP*, *DOCK2*, *CDKN2A*, *ARID1A*, *SMAD4*, *FBXW7*, *KRAS*, *TP53*	Whole-genome sequencing
[1]	TCGA Research Network	2017	*TP53*, *CDKN2A*, *ARID1A*, *SMAD4*, *ERBB2*, *VEGFA*, *GATA6*, *CCNE1*, *KRAS*, *EGFR*, *IGF1R*, *VEGFA*, *GATA4*, *ARID1A*, *SMARCA4*, *PBRM1*	Whole-exome sequencing
[123]	M. Secrier et al.	2016	*SMYD3*, *RUNX1*, *CTNNA3*, *RBFOX1*, *AGBL4*, *INK4/ARF*, *SAMD5*, *CDK14*, *KIF26B*, *THADA*, *SASH1*, *MECOM*, *JUP*, *IKZF3*, *FHIT*, *WWOX*, *MACROD2*, *IMMP2L*, *CCSER1*, *PDE4D*, *NAALADL2*, *PARK2*, *PARD3B*, *PRKG1*, *TP53*, *SMAD4*, *ARID1A*, *CDKN2A*, *KCNQ3*, *CCDC102B*, *CYP7B1*	Whole-genome sequencing
[106]	J. Weaver et al.	2014	*ABCB1*, *ARID1A*, *CCDC102B*, *CCDC153*, *CDKN2A*, *CNTNAP5*, *FGF10*, *MMP16*, *MYD88*, *MYF6*, *MYO18B*, *PCDH9*, *PNLIPRP3*, *SEMA5A*, *SMAD4*, *SMARCA4*, *SSTR4*, *TLR1*, *TLR4*, *TLR7*, *TLR9*, *TP53*, *TRAF3*, *TRAF6*, *TRIM58*, *UNC13C*	Whole-genome sequencing
[105]	A. Dulak et al.	2013	*TP53*, *CDKN2A*, *EYS*, *ARID1A*, *SMAD4*, *PIK3CA*, *SLC39A12*, *SPG20*, *DOCK2*, *AKAP6*, *TLL1*, *TLR4*, *ARID2*, *HECW1*, *ELMO1*, *SYNE1*, *SMARCA4*, *AJAP1*, *C6orf118*, *ACTL7B*, *F5*, *KCNU1*, *NUAK1*, *MYST3*, *SCN10A*, *CNTNAP5*	Whole-exome sequencing

**Table 3 cancers-15-05410-t003:** Genes with CNVs in EAC as suggested by original research.

Reference	Authors	Years	Genes with CNVs	Detected Method
[129]	Sihag et al.	2021	*ERBB2*, *KRAS*, *CCNE1*, *MYC*, *CCND1*, *MDM2*, *VEGFA*, *EGFR*, *CDK6*, *CCND3*, *CDKN2A/B*, *SMAD4*, *ARID1A*, *PIK3CA*, *APC*, *TP53 NGS*	Next-generation sequencing
[120]	Frankell et al.	2019	*ERBB2*, *KRAS*, *SMAD4*	Whole-genome sequencing
[1]	TCGA Research Network	2017	*VEGFA*, *ERBB2*, *SMAD4*, *GATA6*, *CCNE1*	Whole-genome sequencing
[123]	Secrier et al.	2016	50 genes: *CCND1*, *EGFR*, *ERBB2*, *VEGFA*, *KRAS* etc.	Whole-genome sequencing
[107]	Ross-Innes et al.	2015	35 genes: *GATA4*, *KLF5*, *MYB*, *PRKCI*, *CCND1*, *FGF3*, *FGF4*, *FGF19*, *VEGFA*, *A2BP1*, *CDKN2A*, *PDE4D*, *PTPRD*, *PARK2* etc.	Whole-genome sequencing
[108]	Nones et al.	2014	210 genes: *CCNE1*, *ERBB2*, *FRS2*, *GATA4*, *KRAS*, *MTMR9*, *MDM2*, *CDKN2A*, *FHIT*, *RUNX1* etc.	Whole-genome sequencing
[130]	Paulson et al.	2009	47 genes: *EGFR*, *MYC*, *EGFR*, *MTAP*, *CDKN2A*, *CDKN2B*, *SMAD2*, *SMAD4*, *SMAD7* etc.	BAC array CGH
[128]	Pasello et al.	2009	97 genes: *VEGF*, *PTK2*, *ING1*, *SCYA3*, *ABCG2*, *DCC* etc.	MLPA

CNVs: copy number variations; BAC: bacterial artificial chromosome; CGH: comparative genomic hybridization; MLPA: multiplex ligation-dependent probe amplification.

**Table 4 cancers-15-05410-t004:** Genes with DNA methylation in BE and EAC as suggested by original research.

Reference	Authors	Year	Genes	Status	Type of Lesion
[145]	Xu et al.	2013	20 genes: *SH3GL3*, *GBX2*, *SLC18A3*, *SLC6A2* etc.	hypermethylated	BE (vs. NE)
			20 genes: *ZNF625*, *PTPRT*, *ST6GAL2*, *SLC18A3* etc.	hypermethylated	EAC (vs. NE)
[150]	Alvi et al.	2012	*SLC22A18*, *PIGR*, *GJA12*, *RIN2*, *RGN*, *TCEAL7*	hypermethylated	EAC (vs. BE)
[149]	Peng et al.	2011	*MT3*	hypermethylated	EAC (vs. NE)
[146]	Smith et al.	2008	*APC*, *CDKN2A*, *ID4*, *MGMT*, *RBP1*, *RUNX3*, *SFRP1*, *TIMP3*, *TMEFF2*	hypermethylated	EAC (vs. NE)
			*APC*, *ID4*, *MGMT*, *RUNX3*, *SFRP1*, *TIMP3*, *TMEFF2*	hypermethylated	BE (vs. NE)
[147]	Klump et al.	1998	*p16*	hypermethylated	BE (vs. NE)
[148]	Wong et al.	1997	*p16*	hypermethylated	EAC/BE (vs. NE)

BE: Barrett’s esophagus; EAC: esophageal adenocarcinoma; NE: normal esophagus.

**Table 5 cancers-15-05410-t005:** Altered microRNA expression during the development and progression of EAC.

Reference	Authors	Year	MicroRNA	Status	Type of Lesion
[84]	Neureiter et al.	2020	miR-205	upregulated	LE-EAC (vs. RI-EAC)
[161]	Mari et al.	2018	miR125a-5p	downregulated	EAC (vs. NE)
[153]	Kailasam et al.	2015	miR-28, miR-30a-5p, miR-126, miR-143, miR-145, miR-181a/b, miR-199a	upregulated	EAC (vs. NE)
			miR-27b, miR-99a, miR-149, miR-193a/b, miR-210, miR-345, miR-494, miR-513, miR-617, let-7a/b/c	downregulated	EAC (vs. NE)
[152]	Huang et al.	2014	miR-126, miR-143, miR-145, miR-181a, miR-181b, miR-199a, miR-28, miR-30a-5p	upregulated	EAC (vs. HGD)
			miR-149, miR-203, miR-210, miR-27b, miR-513, miR-617, miR-99a let-7a/b/c, miR-193a, miR-345, miR-494	downregulated	EAC (vs. HGD)
			miR-25, miR-93, miR-106b, miR-192	upregulated	EAC (vs. BE)
			miR-203, let-7	downregulated	EAC (vs. BE)
			miR-200a, miR-513, miR-125b, miR-101, miR-197	upregulated	HGD (vs. LGD)
			miR-23b, miR-20b, miR-181b, miR-203, miR-193b, miR-636	downregulated	HGD (vs. LGD)
[154]	Saad et al.	2013	miR-194, miR-31, miR-192, miR-200a	upregulated	EAC (vs. BE)
			miR-203, miR-205	downregulated	EAC (vs. BE)
			miR-194, miR-192, miR-21, miR-31	upregulated	HGD (vs. BE)
[155]	Chen et al.	2013	miRNA-21, miRNA-133b, miR-200a	upregulated	EAC (vs. NE)
[151]	Fassan et al.	2011	miR-215, miR-560, miR-615-3p, miR-192, miR-326, miR-147	upregulated	BE (vs. NE)
			miR-100, miR-23a, miR-605, miR-99a, miR-205, let-7c, miR-203	downregulated	BE (vs. NE)

BE: Barrett’s esophagus; EAC: esophageal adenocarcinoma; NE: normal esophagus; LGD: low-grade dysplasia; HGD: high-grade dysplasia; LE-EAC: localized and early esophageal adenocarcinoma; RI-EAC: regional and invasive esophageal adenocarcinoma.

**Table 6 cancers-15-05410-t006:** Currently employed targeted therapies and immunotherapies for EAC.

Therapy	Target	Patients	Treatment	Approval	Clinical Trials	Current Status
Targeted therapy	HER2	HER2 positive metastatic EAC	Trastuzumab plus chemotherapy	FDA	Phase III (NCT01041404) [15]	In clinic
Targeted therapy	HER2	HER2 positive metastatic EAC	Fam-Trastuzumab Deruxtecan-nxki	FDA	Phase II (NCT03329690) [197]	In clinic
Monotherapy/ Combination therapy	VEGFR2	advanced or metastatic EAC	Ramucirumab (plus paclitaxel)	FDA	Phase III (NCT00917384; NCT01170663) [198,199]	In clinic
Immunotherapy	PD-1	resectable EAC	Nivolumab following nCRT plus radical resection	FDA; EMA	Phase III (NCT02743494) [200]	In clinic
Immunotherapy	PD-1	unresectable HER2 negative metastatic EAC	Nivolumab plus chemotherapy	FDA; EMA	Phase III (NCT02872116) [201]	In clinic
Immunotherapy	PD-1	unresectable HER2 positive metastatic EAC	Pembrolizumab plus Trastuzumab plus chemotherapy	FDA	Phase III (NCT03615326) [202]	In clinic
Immunotherapy	PD-1	locally advanced or metastatic EAC	Pembrolizumab plus chemotherapy	FDA	Phase III (NCT03189719) [194]	In clinic

HER2: human epidermal growth factor receptor 2; VEGFR2: vascular endothelial growth factor receptor 2; PD-1: programmed cell death protein 1; nCRT: neoadjuvant chemoradiotherapy; FDA: Food and Drug Administration; EMA: European Medicines Agency.

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
