# Peer review of "Molecular Biology and Clinical Management of Esophageal Adenocarcinoma"

_cancers, 2023, doi:10.3390/cancers15225410_

Round 1

Reviewer 1 Report

Comments and Suggestions for Authors

The manuscript is interesting and quite well written. I suggest some changes to improve the manuscript, as below reported:

1- Abstract. Therefore, targeting specific cytokines, chemokines, lymphocytes, DNA methylation and microRNAs is highly promising for improving treatment outcomes. Please, underline the novelty of the study.

2- Introduction. L82-86. In this review, we provide a deep insight into the molecular biology of EAC, including etiology, genomic and epigenetic features, the immune microenvironment and histopathology. Furthermore, we discuss the suitability of currently employed strategies for  prevention and treatment of EAC. We aim to provide leads to find more new therapeutics for EAC through this evolving understanding of the molecular biology of EAC. Please, improve the description of study aim.

3- L223-5. In addition, the chemokines, chemokine receptors and cytokines CXCR1, CXCR2, CXCL1, CXCL2, CXCL3, CXCL6, CXCL8, CCL15, CCR4,  IL-2, IL-6, IL-7, IL-8, IL-15, IL-18 are significantly higher expressed in HGD/EAC com-  pared to LGD/BE [91]. Please, also add the term "interleukins".

4- 4. L 517-9. Discussion and conclusion  In this review, we comprehensively discuss the biological features of EAC and BE in  terms of epidemiology, etiology, histology, biology, immunology, genetics and epigenetics. EAC has been a health challenge in Western countries due to the increasing incidence 519 rate. Please, improve the summary of study.

5- Paying more attention to mediators in the tumor microenvironment, such 577 as specific cytokines, chemokines, and lymphocytes, and genetic targets on the level of 578 DNA methylation and microRNAs, may improve treatment outcomes.

6- Please underline the novelty of your study and the possible clinical implications of the analysis carried out in this review. Furthermore, I suggest that the authors propose any unmet needs to be addressed in new papers. 

Comments on the Quality of English Language

Minor changes of English language are required

Author Response

Dear Reviewer,

Kind regards,

Authors

Reviewer 2 Report

Comments and Suggestions for Authors

Specific comments to the authors

The submitted review "Molecular Biology and Advancement of Therapy in Esophageal Adenocarcinoma" collects, summarises and analyses heterogeneous aspects of the aetiology, genetics, epigenetics and histology of esophageal adenocarcinoma (EAC) as well as therapeutic strategies based on previously published reviews as well as in vitro/in vivo experiments and clinical trials.

The topics presented range from classical epidemiology, clinicopathological findings/characteristics and treatment concepts of EAC to important genetics/epigenetics aspects now and in the future. In conclusion, the author gives an interesting overview of genetics and epigenetics related therapy in EAC, which is mostly easy to read, follow and understand. The authors should clarify some aspects before accepting the manuscript for publication, as mentioned below.

Overall, the author could summarise all important clinicopathological as well as genetic and epigenetic findings in an additional figure for better reading, understanding and following, as the presentation so far is largely narrative. In addition, each chapter could include a summary with a final open question at the end.

# Title: The title does not cover all the aspects collected, summarised and discussed in this submitted review.

# 1. Introduction: Please add more details to the sentence "...a small proportion of BE patients may rapidly develop EAC [11]". Please omit speculation such as "We speculate that specific cellular and molecular features confer aggressive behavior and resistance to therapies for EAC. The sentence "The development of other personalized therapies has been hampered by the relatively poor understanding of the cellular and molecular characteristics of EAC" is largely unspecific.

# 2.4 Genetics: Please also add a separate table for copy number alterations.

# 2.5. Epigenetics: Please add a separate table for the findings of DNA methylation and miRNAs in BE and EAC. Please add findings on miRNA, EMT and prognosis (see PMID: 320932609)

# Table 3: Please add the current status and clinical trial ID numbers to the table.

# "4. Discussion: This chapter is largely repetitive and narrative. Therefore, the authors should briefly summarise the most important recent findings in genetic and epigenetic research in EAC. Relevant, predictive and prognostic issues of EAC should be presented and discussed. Finally, the authors should mention some kind of milestones for the improvement of diagnosis and therapy in the future.

Comments on the Quality of English Language

 Minor editing of English language required.

Author Response

Dear Reviewer,

Kind regards,

Authors

Round 2

Reviewer 1 Report

Comments and Suggestions for Authors

All issues raised in the first review have been addressed

Comments on the Quality of English Language

Minor changes of English language are required